# Providing context: Extracting non-linear and dynamic temporal motifs from brain activity

**Eloy Geenjaar** [1,2]*, **Donghyun Kim**[2], **Vince Calhoun**[1,2]

**1** School of Electrical and Computer Engineering, Georgia Institute of Technology, Atlanta, Georgia, United States of America, **2** Tri-Institutional Center for Translational Research in Neuroimaging and Data Science (TReNDS), Georgia State, Georgia Tech, Emory, Atlanta, Georgia, United States of America

* egeenjaar@gatech.edu

**Data availability statement:** The data underlying this study are from the fBIRN phase III study, described in Keator et al. (2015) at

## Abstract

Approaches studying the dynamics of resting-state functional magnetic resonance imaging (rs-fMRI) activity often focus on time-resolved functional connectivity (tr-FC). While many tr-FC approaches have been proposed, most are linear approaches, e.g. computing the linear correlation at a timestep or within a window. In this work, we propose to use a generative non-linear deep learning model, a disentangled variational autoencoder (DSVAE), that factorizes out window-specific (context) information from timestep-specific (local) information. This has the advantage of allowing our model to capture differences at multiple temporal scales. We find that by separating out temporal scales our model's window-specific embeddings, or as we refer to them, context embeddings, more accurately separate windows from schizophrenia patients and control subjects than baseline models and the standard tr-FC approach in a low-dimensional space. Moreover, we find that for individuals with schizophrenia, our model's context embedding space is significantly correlated with both age and symptom severity. Interestingly, patients appear to spend more time in three clusters, one closer to controls which shows increased visual-sensorimotor, cerebellar-subcortical, and reduced cerebellar-visual functional network connectivity (FNC), an intermediate station showing increased subcortical-sensorimotor FNC, and one that shows decreased visual-sensorimotor, decreased subcortical-sensorimotor, and increased visual-subcortical domains. We verify that our model captures features that are complementary to - but not the same as - standard tr-FC features. Our model can thus help broaden the neuroimaging toolset in analyzing fMRI dynamics and shows potential as an approach for finding psychiatric links that are more sensitive to individual and group characteristics.

## Introduction

Complex dynamical systems like the brain often modulate internal representations across various timescales [1,2]. For example, the complete cognitive process of thinking about dinner and the activity generated by populations of neurons effectuating the cognitive process occur

https://doi.org/10.1016/j.neuroimage.2015.09.003, and were collected over a decade ago without a resharing statement, as such the IRB has not approved data sharing outside the study team. Researchers can contact Dr. Theo van Erp for data access requests.

**Funding:** The research presented in this study was supported by the National Science Foundation under Grant No. 2112455 and No. 2316421, as well as by the National Institutes of Health grant #R01MH123610. Eloy Geenjaar was supported by the Georgia Tech/Emory National Institute of Health/National Institute of Biomedical Imaging and Bioengineering Training Program in Computational Neural Engineering (T32EB025816). The funders had no role in the study design, data collection and analysis, decision to publish, or preparation of the manuscript.

**Competing interests:** The authors have declared that no competing interests exist.

at vastly different timescales and represent slowly varying representations (coarse-grained) to fast-varying (fine-grained) representations. Disentangling a timeseries into slowly-varying and fast-varying representations is valuable in a variety of scientific fields, such as video representation learning [3], and healthcare (bio-)signals [4]. It is known that the brain also exhibits a variety of intrinsic neural timescales both during task and resting-state [1] and that these intrinsic neural timescales differ for people diagnosed with schizophrenia [5,6]. In previous work, it has been shown that representations throughout this temporal hierarchy can be modeled based on sensory inputs [2]. However, separating representations at different timescales is harder when sensory inputs are not directly observable, occur at a higher frequency than the imaging modality, or are too complex to accurately encode. These cases require data-driven approaches that can separate slowly-varying from fast-varying representations. Especially given the differences in neuronal timescales for psychiatric subpopulations and the temporal resolution of fMRI data, developing approaches that can infer how representations change over different timescales can aid the discovery of dynamic motifs related to psychiatric disorders such as schizophrenia.

Finding slowly changing representations from neuroimaging data has mostly been explored through functional connectivity (i.e., temporal coherence among isolated regions) or functional network connectivity (FNC) (i.e., temporal coherence among overlapping networks) of the brain [7,8]. These methods have been extended to capture dynamic changes [9–11]. One widely used approach, which uses sliding windows to estimate the functional connectivity of the brain in specific windows [12–14], has been used to show dynamic properties that are linked to schizophrenia [15]. Although temporal coherence or correlation is an interpretable way of representing a window of brain activity, it is also limiting, since there may be more complete ways of summarizing activity in a particular window. Using the framework of separating slowly-varying from fast-varying representations, we can generalize the idea of representing a window with a single embedding to more abstract embeddings. We call these more abstract representations context motifs since they provide context about the brain activity in that window. Not only do these context representations contain information about longer time periods, but the windowing also acts as a low-pass filter that potentially helps denoise some of the underlying information contained in the fMRI data, especially given the low-frequency nature of the BOLD signal, and the relationship between its low-frequency signal and schizophrenia [16].

Our work shows that separately modeling the individual timesteps (local information) and the window as a whole (context information) leads to window embeddings that are more linearly separated between schizophrenia patients and controls. We assess the reliability of our model across different random seeds and find that the reliability between the two main models we propose differs based on their configuration. Mainly, the models are computationally reliable, where the computational reliability of models with fewer context dimensions is more reliable than models with more context dimensions. Then, to understand how our embedding space abstracts away from a functional embedding space, we compare distances in a functional connectivity embedding space with distances in our proposed model's embedding space. Although there is some similarity between the information functional connectivity represents about a window and our context embeddings, we find that the representation spaces are largely different. This means that our model provides complementary and novel information about a window's dynamics that can be used together with FNC as quantitative information in clinical decision making. Lastly, a deeper analysis of a model that is computationally reliable and easy to visualize shows three main clusters of schizophrenia patients in the context space. One cluster with windows closer to control subject windows, one middle cluster that slightly overlaps with control subjects, and one cluster that is

mostly completely separated from control subjects. To understand what these clusters represent, we visualize them as connectivity patterns and find interesting patterns relating to visual-sensorimotor, subcortical-cerebellar, subcortical-sensorimotor, and cerebellar-visual. Specifically, the windows in the separated cluster, including reduced visual-sensorimotor and increased subcortical-sensorimotor connectivity, represent unique motifs that are seen in these most separated windows from schizophrenia patients, thus providing a powerful and more fine-grained way to identify functional patterns that are linked to the disorder.

## Materials and methods

### Related work

To our knowledge, there are no methods that explicitly try to disentangle local and context representations from brain data. Apart from approaches based on sliding windowed Pearson correlation, which is commonly used to derive brain states (e.g., dFC/dFNC), the most similar model to ours uses a 1D convolutional autoencoder to learn embeddings for windows of fMRI activity [17]. Our work extends this model by learning embeddings for both the window and the individual timesteps in the window, and the interaction between the context and individual timestep embeddings. This allows us to separate local information from the context embeddings. Other relevant work finds component-specific temporal primitives with a self-supervised learning framework [18]. We generalize this work towards spatio-temporal patterns that span multiple components, and because our model contains a decoder, we can more easily interpret the embedding space our model learns. Our work is inspired by progress in adjacent fields [4], and we adapt both the factorized (independent) and unfactorized version of the disentangled sequential autoencoder (IDSVAE and DSVAE, respectively) [3] to brain data.

### Model definition

Let $x_{1:T} \in \mathbb{R}^{N \times T}$ denote the multivariate timeseries of a single subject, with $T$ the number of timesteps and $N$ the number of features. In our case, the features are independent component analysis (ICA) component time courses. Our goal is to learn a generative model that can be generalized to new subjects in a test set. The way we factorize our generative model is to separate local information for each timestep, and context information for each window. A graphical representation of this factorization is shown in Fig 1.

To achieve this factorization, we first separate the dataset into (overlapping) windows with a window size of 32 timesteps, which is equivalent to 64 seconds, and 4 steps between each window, equivalent to 8 seconds. For a window of data $x_{w_j}$, where $j$ indexes over each window, we learn two separate encoders. One context encoder, $\phi_\theta^{\text{context}}(\cdot)$, and a local encoder $\phi_\theta^{\text{local}}(\cdot)$. These encoders have learnable parameters $\theta$, and the context encoder takes the full window as input $z_c = \phi_\theta^{\text{context}}(x_{w_j})$, where we refer to $z_c$ as a context embedding. For the DSVAE model, we concatenate this context vector $z_c$ with the window $x_{w_j}$ to form the input to the local encoder: $z_{1:W} = \phi_\theta^{\text{local}}([z_c x_{1:W}])$. For the unfactorized version of the DSVAE [3] (IDSVAE) that we also use in this paper, the input into the local encoder is just the window of fMRI data $x_{w_j}$. The only two differences between the DSVAE and IDSVAE are that in the DSVAE, the context embedding is used to compute the local embeddings and the prior for the local embeddings in that window. The final step in the model is the spatial decoder $\psi_\theta(\cdot)$, a multilayer perceptron (MLP), that takes a concatenated vector of the context embedding and the local embedding $\hat{x}_t = \psi_\theta([z_c z_t])$ to reconstruct the original timestep from the fMRI data $x_t$. Since the context embedding is learned for a specific window, it remains the same for

### 1) rs-fMRI timeseries

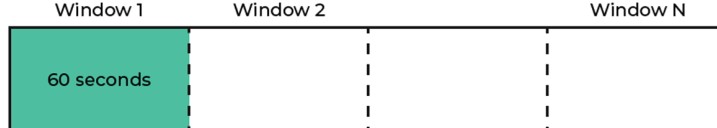

### 2) Local context disentangling (DSVAE)

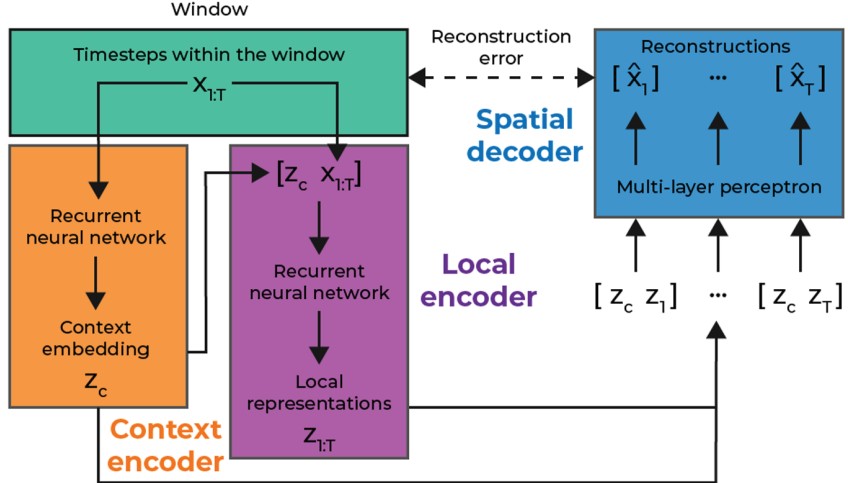

**Fig 1. An abstract depiction of our model, for the independent version of the model, the context representation is not concatenated inside the local encoder.**

all timesteps in a window. We train the model using a reconstruction loss, and a variational loss [19], which acts as regularization on the context and local embeddings, as follows.

$$\frac{1}{M}\sum_j^M \mathcal{L}(\phi_\theta^{\text{context}}, \phi_\theta^{\text{local}}, x_{w_j}) = \frac{1}{W}\sum_t^W \mathbb{E}_{z_c, z_t}\left[\log \psi_\theta\left([z_c, z_t]\right]\right] +$$

$$\beta\frac{1}{W}\sum_t^W D_{\text{KL}}\left(\phi_\theta^{\text{local}}(x_t) \| p(z^{\text{local}})\right) +$$

$$\gamma D_{\text{KL}}\left(\phi_\theta^{\text{context}}(x_{w_j}) \| p(z^{\text{context}})\right) \tag{1}$$

where $\beta$ and $\gamma$ are hyperparameters that weigh the regularization terms in the total loss. Note that the KL-divergence $D_{\text{KL}}(\cdot)$ pushes the distributions parameterized by the context and local encoder to be close to their respective priors $p(z^{\text{context}})$ and $p(z^{\text{local}})$. The context prior is a zero-mean unit-variance normal distribution, whereas the local prior is a learnable GRU that is conditioned on the context embedding for the DSVAE, and conditioned on a learnable vector for the IDSVAE.

## Data and resources

In this work, we use the function bioinformatics research network (fBIRN) phase III data, with over 300 schizophrenia patients and controls [20]. The data was first accessed for this project on September 25, 2023. All subjects were unidentifiable to the authors. The demographics for control subjects and schizophrenia patients are described in Table 1. The

**Table 1. Data sample demographics. Note that AP refers to anti-psychotic medication, and AD refers to anti-depressive medication. Thus, AP and AD in this table refer to the percentage of patients taking anti-psychotic and anti-depressive medication, respectively. PANSS is a symptom scale for schizophrenia. We show its positive, negative, and composite scores.**

|  | Schizophrenia patients | Control subjects |
|---|---|---|
| N subjects | 151 | 160 |
| Female (%) | 23.84 | 23.13 |
| Avg Age | 39+-12 | 37+-11 |
| Avg Psychosis length | 17+-11 | - |
| AP (%) | 88.74 | - |
| AD (%) | 37.09 | - |
| Avg AP length | 15+-10 | - |
| CMINDs | -1.6+-1.2 | 0.0+-0.9 |
| PANSS positive | 15.3+-5.0 | - |
| PANSS negative | 14.3+-5.6 | - |
| PANSS composite | -1.0+-6.3 | - |

schizophrenia patients and controls were matched based on age, gender, handedness, and race distributions.

The dataset is collected at seven consortium sites (University of Minnesota, University of Iowa, University of New Mexico, University of North Carolina, University of California Los Angeles, University of California Irvine, and University of California San Francisco). Each of these consortiums records diagnosis, age at the time of the scan, gender, illness duration, symptom scores, and current medication, when available. The inclusion criteria were that participants were between 18 and 65 years of age, and their schizophrenia diagnosis had to be confirmed by trained raters using the Structured Clinical Interview for DSM-IV (SCID) [21]. All participants with a schizophrenia diagnosis were on a stable dose of antipsychotic medication; either typical, atypical, or a combination for at least two months. Each participant with a schizophrenia diagnosis was clinically stable at the time of the scan. The control subjects were excluded based on current or past psychiatric illness or in case a first-degree relative had an Axis-I psychotic disorder. These diagnoses were based on the SCID assessment. Written informed consent from all study participants was obtained under protocols approved by the Institutional Review Boards at each consortium site.

As a preprocessing step, we obtain ICA timeseries from the rs-fMRI data using the fully automated NeuroMark pipeline [22]. Specifically, we implement NeuroMark, using the NeuroMark_fMRI_1.0 template (the template is released in the GIFT software at

http://trendscenter.org/software/gift and results in 53 ICA components and timecourses from each subject. Given the robustness of the NeuroMark pre-processing pipeline to short scans [23], we do not test how pre-processing differences affect our model, but instead assess the effect of training-time noise in S2 Appendix. We find essentially no deterioration in classification accuracy with training-time noise, and that the DSVAE model has better reliability under training-time noise than the IDSVAE model.

The necessary code to reproduce the proposed model is available on GitHub. Each model is trained with PyTorch 2.0 [24] across 32 different hyperparameters with 4 seeds [42,1337,1212,9999] (128 runs total), and the hyperparameters with the best validation performance on average across the seeds is used in the final evaluation of each model. The hyperparameter ranges for each model are described in S1 Appendix. Each model is trained with an NVIDIA GeForce RTX 1080 or A40. A comparison between a recurrent neural network (RNN) and convolutional context encoder is shown in S4 Appendix

## Experiments

We have devised a series of experiments to verify that our method can find embeddings that are distinctly different from wFNC representations and are clinically relevant. First, in Section Window classification, we verify whether embeddings obtained with our method are linearly separable. We should be able to classify whether a window of fMRI activity is from a schizophrenia patient or a control subject. Second, in Section Reliability analysis, we evaluate how reliable each method is across initialization seeds. Models with similar embeddings across initialization are more robust and thus more impactful. Clinical inferences can not be made based on models that depend too much on the random seed because it is hard to decide which random seed is the 'correct' model to derive inferences from. Third, in Section Cluster analysis, we analyze what clusters we find in our model's embedding space and how these clusters relate to demographic variables and cognitive scores. Lastly, in Section Manifold comparisons, we analyze how the wFNC embeddings differ from the embeddings our model produces. One of the goals of our work is to find a novel way to summarize fMRI window dynamics. We therefore want to analyze whether the embeddings are significantly different from wFNC representations by looking at distances between windows in both spaces and understanding how well the distances are matched between our model's embedding space and wFNC representations.

### Window classification

To test how well windows of fMRI activity from schizophrenia patients and control subjects are linearly separable in a latent space, we compare our proposed model to two baseline models as well as the widely used wFNC approach. The two baseline models we use are a model that only uses local embeddings (LVAE), and a model that only uses context embeddings (CVAE). The latter model has been proposed in previous work [17], and we follow their design choices by using a convolutional encoder and decoder. The LVAE model is the same as our model, but only uses local embeddings and no context embeddings. To obtain context embeddings from the LVAE model, we average the local embeddings over time in a window. To obtain embeddings of different sizes for wFNC, we use PCA on the training and validation set and then transform the test set into the same PCA space. To perform the classification, we concatenate the context embeddings from the training and validation set and train a linear support vector machine to separate context embeddings from schizophrenia subjects and control subjects in the latent space. The classification accuracy is calculated using the context embeddings from subjects in the test set. The reason we specifically chose to use a linear evaluation is because the latent space of variational autoencoders (VAEs) is often assumed to be locally Euclidean. This enables linear interpolation between points in the latent space to be meaningful when decoded into the original data space. Since our models are trained with a variational loss, and because linear separation makes our exploration of group differences more interpretable in Sections Cluster analysis & Cluster visualization, we decided to focus on linear separability. The results are shown in Fig 2. Note, we compare models with different local and context sizes to verify results are not spurious with respect to the chosen local and context embedding sizes.

Based on Fig 2, we can conclude that both the independent (IDSVAE) and dependent (DSVAE) versions of our model outperform all other methods. Compared to the LVAE and CVAE baseline, it is important to separate local and context information to obtain embeddings that separate windows of activity from schizophrenia subjects and control subjects. Moreover, although wFNC performs better than the LVAE and CVAE baselines, our proposed methods outperform wFNC. It is hard to generally say whether independence (IDSVAE) or

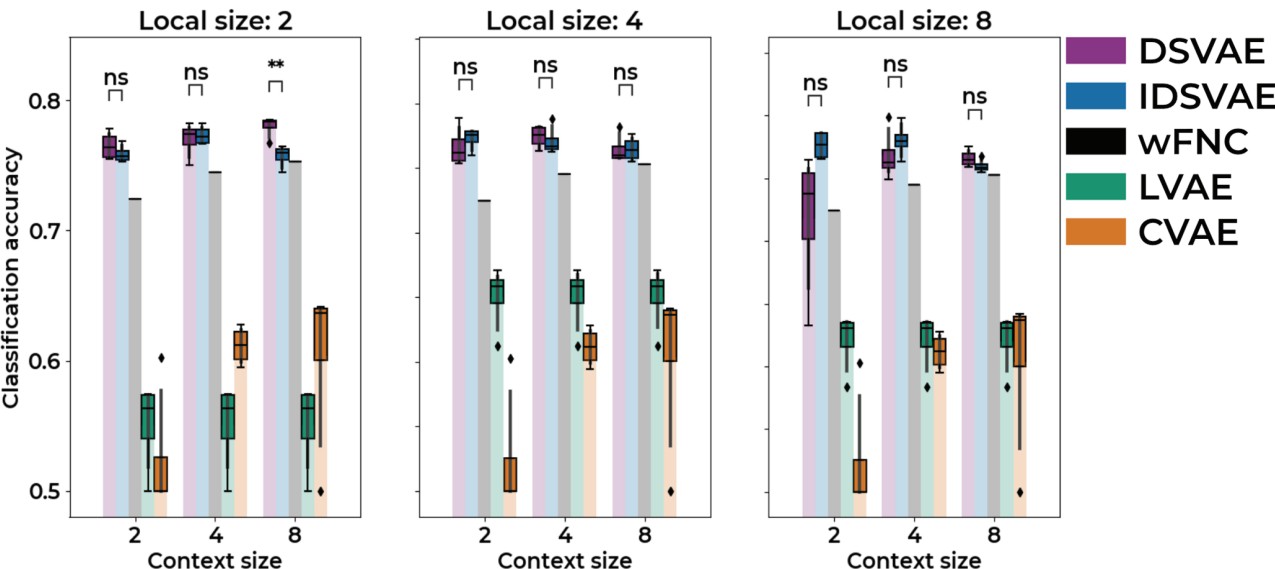

**Fig 2. The window classification accuracy for two of our proposed models (DSVAE, IDSVAE), windowed FNC (wFNC), and two baseline methods: context-only (CVAE), and local-only (LVAE).** Our proposed methods outperform all other methods, experiments are performed across 4 seeds. Significance levels of the independent t-test: *: $p<0.05$, **: $p<0.01$.

dependence (DSVAE) is better for the classification accuracy of our method. Only for local size = 2, and context size = 8 is DSVAE significantly better ($p<0.01$). For all other combinations of local and context sizes, the models do not significantly outperform each other.

## Reliability analysis

It is important for neural network models to converge to the same solution, especially if we want to use them to make clinical inferences. To test whether different instantiations of our DSVAE and IDSVAE models converge to the same embedding space, we perform a reliability analysis. In essence, we are testing the computational reproducibility of the method, but across different random seeds. The results of this reliability analysis are shown in Fig 3. To test across-seed reliability, we embed the full dataset (training, validation, and test set) for each model and each of the four seeds. Then, for each combination of seeds, we fit a linear regression model to predict the location of each context embedding in another seed's embedding space, on the concatenated training and validation set. We then use the average R-squared score across each combination of seeds as a metric of similarity across seeds. Namely, if the context embedding space is the same under linear transformations across seeds, we believe that the model converges to a similar solution irrespective of the initialization of the network.

In Fig 3 we can see two main trends. First, that the DSVAE model performs the best when the local size and context sizes are smallest: (LS=2,CS=2 $p<0.001$, LS=2,CS=4 $p<0.0001$, and LS=4,CS=2 $p<0.05$), and that the IDSVAE is more reliable with a larger local size (LS=8,CS=2 $p<0.001$ and LS=8,CS=8 $p<0.0001$). Moreover, larger context dimensionalities generally lead to worse reliability. This is largely expected since higher-dimensional spaces are essentially 'bigger', and there are thus more variations of embedding spaces that can lead to a good generative model of the data. Furthermore, in the cases where the DSVAE model is better than the IDSVAE model, its reliability is fairly high overall. Especially the LS=2,CS=4 model

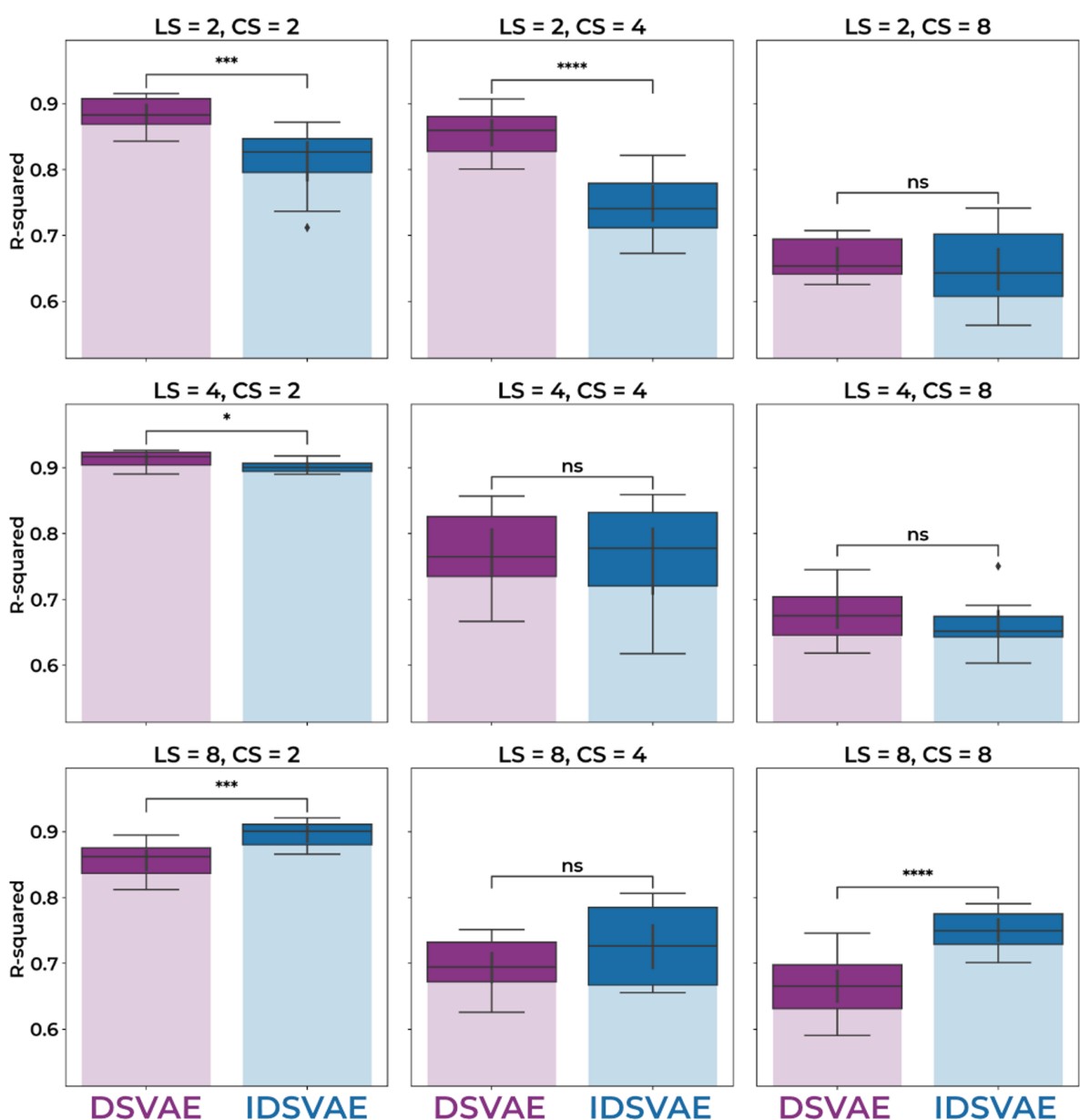

**Fig 3. Each subfigure shows a different local size (LS) and context size (CS) configuration, where the reliability of the model across different initializations is measured in R-squared.** In most cases where both the local size and context size is small, the DSVAE model is significantly more reliable than the IDSVAE method. However, for a local size of 8, the IDSVAE method is significantly more reliable when the context sizes are 2 or 8. Moreover, reliability decreases with a larger dimensionality of the context space, likely increasing the dimensions essentially increases the size and thus the number of equivalent solutions of the space. Significance levels of the independent t-test: *: $p<0.05$, **: $p<0.01$, ***: $p<0.001$, and ****: $p<0.0001$.

both has a high classification accuracy, see Fig 2, good computational reproducibility, see Fig 3, and important for the next sections, is easy to visualize because the context embeddings are 2-dimensional. Hence, we will use this model for further analysis in the upcoming sections.

## Manifold comparisons

Given our model's improved performance over wFNC in Section Window classification, we look at the similarity between our model's embedding space and that of wFNC representations. To formalize this notion, we compare the similarity between normalized distances in the wFNC space and our model's embedding space. First, we embed all of the windows in our model's embedding space and compute their wFNCs. Then, we calculate the distance matrix between the windows in both the wFNC space and our model's embedding space using the Euclidean distance. Lastly, we train a linear regression model to predict distances between windows in our model's embedding space from distances between windows in the wFNC space. The R-squared score is 0.14, this indicates that our model does not learn features that are essentially similar to connectivity features. Our model thus learns unique dynamical features and can provide complementary quantitative information about a window of fMRI data to practitioners. We additionally provide a visualization of wFNC features based on our model's embedding space in S3 Appendix to show differences in representations.

## Cluster analysis

After verifying our model's computational reproducibility, we visualize the embedding space of a highly computationally reproducible version of our model. For the visualization, we use the LS = 4,CS = 2 version of our model. Since the context embeddings are 2-dimensional, we can easily visualize the embedding space of our model, as shown in Fig 4. In the patient-only version of the plot, there seem to be three clusters, which roughly correspond to windows that are more similar to windows from control subjects (bottom), windows that are on the

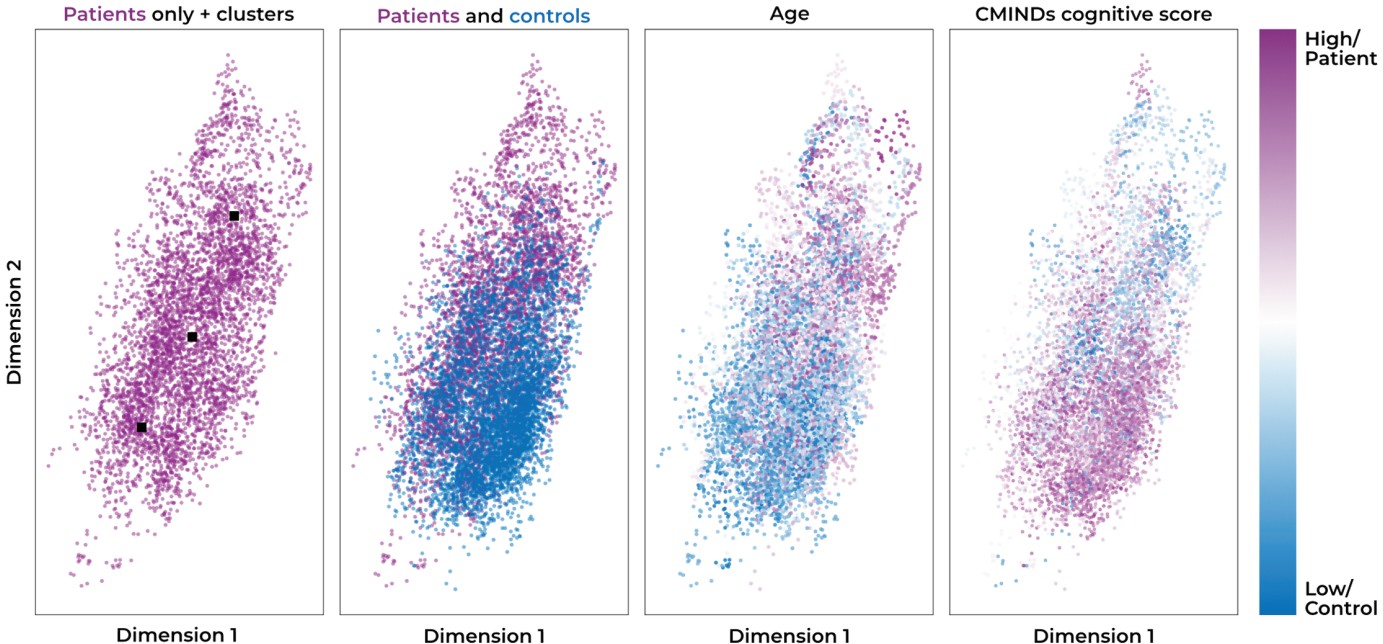

**Fig 4. These subfigures show a visualization of the context embeddings for the DSVAE model with LS=2=4,CS=2.** The first subfigure from the left shows three patient clusters, the second subfigure both patients and controls, the third colors context embeddings based on the subject's age, and the last subfigure colors the context embeddings based on the subject's cognitive score, CMINDs.

boundary (middle), and windows that are completely dissimilar from any control subject windows (top). We obtain the cluster centers using K-means clustering [25]; the clusters are black squares in the leftmost subfigure in Fig 4. The cluster that is most separated from control windows visually seems to represent a cluster of older subjects, with a low CMINDs [26] score, as shown in Fig 4.

To verify the aforementioned visual relationship between the most separated cluster of windows from schizophrenia subjects, and age and CMINDs score, we perform statistical tests. First, we calculate a two-sided t-test between subjects diagnosed with schizophrenia who do have a window that is present in the cluster, and subjects diagnosed with schizophrenia who do not. We find both significant differences for age ($p<5E–6$, $t = 4.96$) and CMINDS score ($p<0.05$, $t = –2.49$. Lastly, we calculate the number of times a window from a subject appears in the cluster and calculate the Pearson correlation between the time spent in the cluster and the age/CMINDS score for schizophrenia patients. This analysis is similar to dwell time analyses [27]. Again, we find significant differences for age ($p<5E–5$, $r = 0.34$) and CMINDs score ($p<0.05$, $r = –0.17$). On the other extreme, there is the cluster of windows from subjects diagnosed with schizophrenia that is most similar to windows from control subjects. We find the opposite effects with the t-test for age ($p<5E–7$, $t = –5.8$) and CMINDs score ($p<0.005$, $t = 3.24$), and Pearson correlation for age ($p<5E–7$, $r = –0.42$) and CMINDs score ($p<0.005$, $r = 0.28$).

## Cluster visualization

To interpret the clusters and the types of motifs they represent in an interpretable format, we visualize the averaged wFNC matrix for each cluster. To find the wFNC matrix for each cluster, we compute the wFNC representation for each point in the embedding space, and average the wFNC's of points that belong to the same cluster. Lastly, to highlight the differences in the clusters, we subtract the average schizophrenia wFNC from each of the clusters. These final wFNC representations of the clusters are shown in Fig 5.

In the Cluster analysis section, we saw that the three clusters essentially encode a gradient from windows that are similar to control windows in Cluster 1, to windows that are not

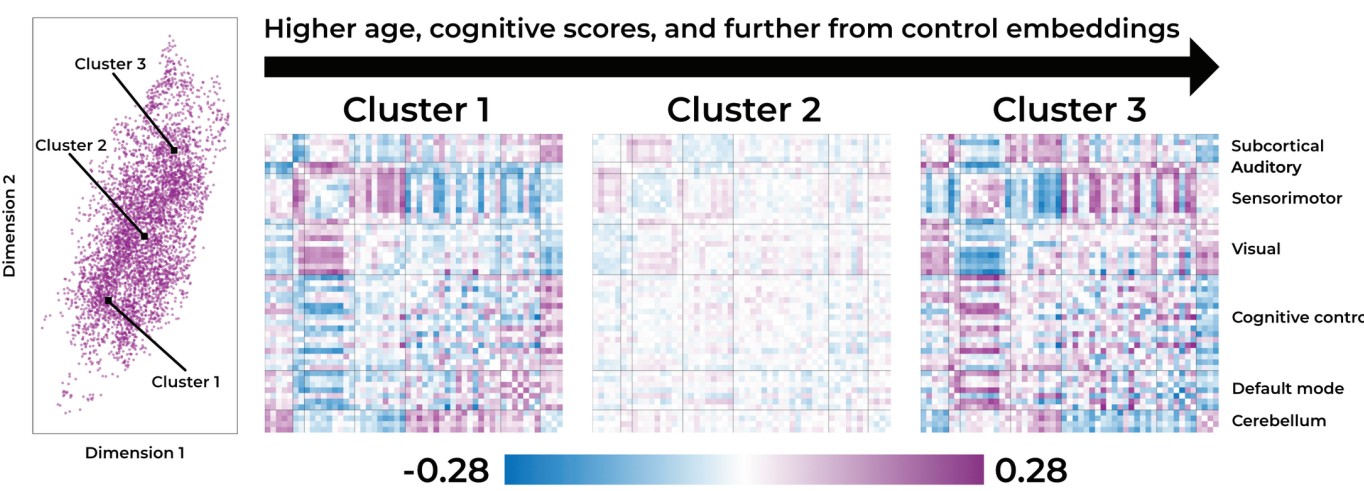

**Fig 5. The three schizophrenia patient clusters visualized using functional connectivity matrices. To create the visualizations, we take all the windows belonging to a cluster and average their wFNC matrix.**

necessarily similar to control windows, but also not easily separable in Cluster 2, and almost completely separated windows from schizophrenia patients in Cluster 3. Cluster 1, which is a cluster closer to controls, shows increased visuo-motor, cerebellar-subcortical, and reduced cerebellar-visual functional network connectivity (FNC). Then, the middle cluster (Cluster 2), is generally less different from the average schizophrenia wFNC than Cluster 1 and 3, but shows increased subcortical-sensorimotor FNC, whereas Cluster 1 and 3 do not. Lastly, the cluster that is most separated from the control windows, shows reduced visuo-sensorimotor, reduced subcortical-sensorimotor, and increased subcortical-visual connectivity.

## Discussion

In this work, we proposed a model that can be used as complementary or an alternative to wFNC. In the Window classification section, we first show that both of our proposed methods are better than baseline models and wFNC in separating windows of rs-fMRI activity from schizophrenia patients. These results indicate that by factoring out timestep-specific and window-specific information, we obtain context embeddings that contain information about the subject's diagnosis. In the Window classification section we hypothesize that factoring out context embeddings potentially helps the model focus on the more low-frequency signal in the rs-fMRI timeseries, potentially de-noising some of the signals. Spontaneous low-frequency fluctuations in the BOLD signal have been linked to schizophrenia [16]. Moreover, it is unlikely that individual timesteps are directly linked to psychiatric disorders. Instead, our context embeddings obtain information from a larger set of timesteps. Since windows capture longer periods of fMRI activity, they are more likely to contain information about cognitive function. Furthermore, the dynamics within a window may be important in recognizing potentially dysfunctional motifs of fMRI activity. These dynamics can only be captured in a window of activity, as opposed to a single timestep. To verify that our results are computationally reproducible, we then calculate how similar embedding spaces across initializations are in the Reliability analysis section. Our results show that the DSVAE model is more often computationally reproducible than the IDSVAE model, especially with smaller local and context sizes. One larger trend we observe for the computational reproducibility results is that larger context sizes lead to less reproducible models. This observation makes sense because higher dimensionality, paired with a KL-divergence regularization, leads to a 'larger' overall space to span. Thus, across random initializations, the model can find 'good' solutions that have different latent spaces, this is a consequence of underspecification [28]. Lastly, especially models with smaller context sizes are computationally reproducible, with R-squared scores around 0.9. Given the computational reproducibility of the DSVAE model with 4-dimensional local and 2-dimensional context embedding spaces, we further analyze its embedding space in subsequent sections.

In the Manifold comparisons section we verify that the features our model learns are unique by comparing our model's embedding space to that of the wFNC's embedding space. We find that our model learns different features than wFNC does. This means our model provides complementary results to wFNC, and can thus help future research in uncovering motifs that are not connectivity-based. This is important because correlations are invariant to scaling, temporal permutation(if the permutation is the same for all inputs), and the addition of a constant, connectivity features from correlations are highly specific. Although connectivity is interpretable, we propose a method that can learn more abstract embeddings that are more indicative of schizophrenia, and potentially other psychiatric diagnoses. As a complementary quantitative tool to wFNC that can summarize dynamical motifs from fMRI data, it can help practitioners make more informed data-driven decisions.

Lastly, in our cluster analysis, we found a cluster of windows from schizophrenia subjects with significantly lower CMINDS scores and higher ages. The relationship between lower CMINDS scores and higher age has been reported previously [26]. A lower CMINDS score indicates higher symptom severity, and in the results presented in the Cluster analysis section, the y-axis captures symptom severity, and the age of the subject. In [26] it was found that the CMINDS score does not exhibit significant group-by-age interactions. It is thus not entirely clear how age and the CMINDS score interact with schizophrenia diagnosis. However, our finding that there is a cluster of windows in our model's embedding space coming from significantly older schizophrenia patients and patients with a significantly worse CMINDS score, indicates that our model is capturing meaningful variation in either age, symptom severity, or both, which likely leads to the increased classification performance in the Window classification section. To interpret how the three clusters differ from each other in terms of the dynamic motifs that they represent, we visualized the corresponding wFNC of each cluster. The three schizophrenia clusters roughly correspond to (1) windows that are similar to control windows, (2) a cluster that is in between schizophrenia patients and control subjects, and (3) a cluster with windows that are almost entirely separated from control subject windows. The first cluster shows increased visual-sensorimotor, cerebellar-subcortical, and reduced cerebellar-visual functional network connectivity (FNC), which are connectivity regions that align with previous connectivity-based results between control subjects and schizophrenia patients [29]. The second cluster shows increased subcortical-sensorimotor FNC, which is intriguing because it is only increased in this cluster, and more generally decreased in the other two clusters, even though the other two clusters are the most dissimilar. These findings align with but extend, previous work [30] that found increased functional connectivity between the thalamus and sensorimotor network for schizophrenia patients with psychomotor excitation, as opposed to decreased connectivity for schizophrenia patients with psychomotor inhibition. Lastly, the most separated cluster shows decreased visual-sensorimotor, decreased subcortical-sensorimotor, and increased visual-subcortical domains. These increases and decreases are with respect to the rest of the schizophrenia patients, and thus do not reflect a decrease or increase with respect to control subjects necessarily. Previous work has shown that reduced visuo-motor could be prospectively related to social deficits in schizophrenia patients [31]. The patterns observed in the most separated cluster in our analysis, which shows reduced visual-sensorimotor connectivity, may thus be indicative of negative symptoms. Additionally, decreased cortical-subcortical motor loop interaction has been related to hypokinesia, a general slowing of movement and a negative symptom, in schizophrenia patients [32,33]. This aligns with the decreased subcortical-sensorimotor connectivity we find in the most separated cluster, and underlines its potential indication as a severity of negative schizophrenia symptoms.

It is important to note that our findings also indicate specific windows from the fMRI timeseries where certain connectivity patterns are either higher or lower with respect to the average wFNC across all subjects diagnosed with schizophrenia. Since our findings indicate that subjects diagnosed with schizophrenia who are older and have lower CMINDs scores spend significantly more time in these connectivity patterns. Especially given the significantly lower CMINDs scores, these dynamic patterns and how often they occur during a resting-fMRI scan indicate an increasing severity of symptoms, and as mentioned above, negative symptoms specifically. Since the identified patterns of connectivity are specific to certain areas of the brain, stimulation in those areas could potentially help disrupt the dynamical patterns we have identified as being linked to schizophrenia. With the effectiveness of transcranial magnetic stimulation (TMS) in treating negative symptoms for patients diagnosed with

schizophrenia [34], our findings indicate potentially fruitful stimulation sites that relate to symptom severity.

### Future work

The model we propose in this work can in future work be expanded to additional datasets and other psychiatric disorders. In its current form, there are no assumptions about the structure of the data, except that it is possible to create temporal windows, and can thus be used for both task and resting-state fMRI data or even EEG data. This also means future work can utilize window size generalizations, such as hierarchical windows or windows that vary in size across the timeseries. It is also important in future work to develop interpretable ways of visualizing the data; although we use wFNC matrices to visualize the clusters in this work, we also find that our model learns new features that are not captured by wFNC, so there are potentially other ways to visualize our model's embedding space. The complementary features our model learns open up a different space in which we can study the rs-fMRI dynamics of psychiatric patients, and potentially enable a deeper understanding of psychiatric disorders, such as schizophrenia. With the heterogeneous nature of certain psychiatric disorders, including schizophrenia, it is important to understand how brain activity differs across a variety of features. In this work, we have started exploring how schizophrenia patients vary across the features our model learns, but we believe our results warrant further clinical research.

### Conclusion

In this work, we tried to propose a more abstract way in which windowed rs-fMRI can be summarized that can complement popular approaches like wFNC. We first show that our proposed models outperform baselines and wFNC in separating windows from schizophrenia patients and control subjects. For both of our proposed models, we assess their computational reliability, which is an important aspect of biomedical machine-learning models. With a computationally reliable and interpretable model, we then dig deeper into how the trained model differs from wFNC in the features that it learns. Moreover, for individuals with schizophrenia, we find that a separated cluster of windows from schizophrenia patients in our model's context embedding space is significantly correlated with both age and symptom severity. In general, patient windows appear to be split into three clusters, one with windows more similar to controls, and with increased visuo-sensorimotor, cerebellar-subcortical, and reduced cerebellar-visual functional network connectivity (FNC). For the station that slightly overlaps with control subjects, we find increased subcortical-sensorimotor FNC, which is not found in the other two clusters and is an interesting result that may indicate schizophrenia patients with psychomotor excitation. The last cluster, which is most separated from control subjects shows decreased visuo-sensorimotor, decreased subcortical-sensorimotor, and increased visuo-subcortical domains. Our proposed model can thus help broaden the neuroimaging toolset in analyzing fMRI dynamics and shows potential as an approach for finding psychiatric links that are more sensitive to individual and group characteristics.

### Supporting information

**S1 Appendix. Hyperparameter ranges**
(ZIP)

**S2 Appendix. The influence of training-time noise on classification accuracy and reliability**
(ZIP)

**S3 Appendix. Geometric manifold comparison visualization**
(ZIP)

**S4 Appendix. Convolutional vs RNN context encoder**
(ZIP)

## Author contributions

**Conceptualization:** Eloy Geenjaar, Donghyun Kim, Vince Calhoun.

**Data curation:** Eloy Geenjaar.

**Formal analysis:** Eloy Geenjaar, Donghyun Kim.

**Funding acquisition:** Vince Calhoun.

**Investigation:** Eloy Geenjaar.

**Methodology:** Eloy Geenjaar, Donghyun Kim.

**Project administration:** Vince Calhoun.

**Resources:** Eloy Geenjaar, Vince Calhoun.

**Software:** Eloy Geenjaar, Donghyun Kim.

**Supervision:** Vince Calhoun.

**Validation:** Eloy Geenjaar.

**Visualization:** Eloy Geenjaar, Donghyun Kim.

**Writing – original draft:** Eloy Geenjaar.

**Writing – review & editing:** Eloy Geenjaar, Vince Calhoun.

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
