## [Decision Letter · Decision Letter 0]

PONE-D-24-29597Providing context: Extracting non-linear and dynamic temporal motifs from brain activityPLOS ONE

Dear Dr. Geenjaar,

Thank you for submitting your manuscript to PLOS ONE. After careful consideration, we feel that it has merit but does not fully meet PLOS ONE’s publication criteria as it currently stands. Therefore, we invite you to submit a revised version of the manuscript that addresses the points raised during the review process.

The manuscript requires minor revisions, primarily related to the interpretation of results. Please address the specific issue raised by Reviewer #2 and revise the discussion in accordance with the reviewers' suggestions.

Reviewer #1 provided some general comments. While the authors are required to respond to these comments, no changes to the manuscript are necessary for the points that can be addressed  in the response to reviewers.

We look forward to receiving your revised manuscript.

Kind regards,

Federico Giove, PhD

Academic Editor

PLOS ONE

Journal Requirements:

“NSF Grant No. 2112455,

NSF Grant No. 2316421

NIH grant #R01MH123610

Eloy Geenjaar was supported by the Georgia Tech/Emory NIH/NIBIB Training Program in Computational Neural-engineering (T32EB025816). “

4. Please expand the acronym “NSF” (as indicated in your financial disclosure) so that it states the name of your funders in full.

“This material is supported by the National Science Foundation under Grant No. 2112455, No. 2316421, and the National Institutes of Health grant #R01MH123610. Eloy Geenjaar was supported by the Georgia Tech/Emory NIH/NIBIB Training Program in Computational Neural-engineering (T32EB025816).”

“NSF Grant No. 2112455,

NSF Grant No. 2316421

NIH grant #R01MH123610

Eloy Geenjaar was supported by the Georgia Tech/Emory NIH/NIBIB Training Program in Computational Neural-engineering (T32EB025816). “

Reviewers' comments:

Reviewer's Responses to Questions

**Comments to the Author**

1. Is the manuscript technically sound, and do the data support the conclusions?

Reviewer #1: Yes

Reviewer #2: Yes

2. Has the statistical analysis been performed appropriately and rigorously? 

Reviewer #1: Yes

Reviewer #2: Yes

3. Have the authors made all data underlying the findings in their manuscript fully available?

Reviewer #1: Yes

Reviewer #2: Yes

4. Is the manuscript presented in an intelligible fashion and written in standard English?

Reviewer #1: Yes

Reviewer #2: Yes

5. Review Comments to the Author

Reviewer #1: Strengths

The paper provides a comprehensive evaluation of multiple embedding methods (IDSVAE, DSVAE, LVAE, CO, and wFNC), offering detailed comparisons across linear separability, reliability, and clinical relevance. This broad scope ensures a well-rounded assessment of the proposed models. By performing a reliability analysis across multiple random seeds, the study tackles the critical issue of reproducibility in neural network performance. Additionally, the inclusion of cluster analysis to examine relationships between embedding clusters and demographic variables, such as age and cognitive scores (CMINDs), links the model’s findings to clinically meaningful insights, enhancing the research's potential impact.

Weaknesses

1.) The paper seems to focus primarily on schizophrenia and control subjects without extensive exploration of other neurological or psychiatric conditions. This limits the generalizability of the findings to broader clinical contexts.

2.) While linear separability is an important metric, the heavy focus on it may overlook other potentially relevant aspects of the embeddings, such as non-linear relationships or other forms of separability that could be clinically significant.

3.) While the study links embedding clusters to demographic variables and cognitive scores, the clinical interpretation of these findings is not fully explored. There is a gap between the technical findings and their practical application in clinical decision-making, which could limit the study’s real-world impact.

4.) The study primarily focuses on varying local and context sizes but does not extensively explore other variations in model architecture or hyperparameters that might influence performance, such as different types of encoders/decoders or alternate methods for embedding aggregation.

Comments to authors

1.) The trade-off between classification accuracy and computational reproducibility is a key issue highlighted in the paper. The IDSVAE model performs better in some cases but has lower reliability, suggesting a need for further refinement to balance these two aspects effectively.

2.) The manifold comparison shows that the proposed model's embeddings do not closely match wFNC features, with a low R-squared score of 0.19. While this indicates that the new embeddings capture different features, the clinical or biological significance of these differences is not well explained.

3.) The visualization of clusters and embeddings in two dimensions is a strength, but the interpretation of these visualizations needs to be expanded. More detailed explanations and additional metrics must be provided to link the visual patterns to specific clinical or cognitive outcomes.

4.) The analysis of distance traveled by embeddings between time steps reveals some differences between schizophrenia patients and controls. However, this observation needs to be discussed in detail to understand the underlying dynamics or to correlate these movements with specific neural or behavioral phenomena.

Reviewer #2: Thanks for inviting me to review the paper by Geenjar et al. " Providing context: Extracting non-linear and dynamic temporal motifs from brain activity ". In this study, the authors propose to use a generative non-linear deep learning model, a disentangled variational autoencoder (DSVAE), to capture differences across different cohorts of subjects at multiple temporal scales.

They find that for individuals with schizophrenia, the largest temporal scale is significantly correlated with both age and symptom severity and more accurately separates chunks of timeseries at larger scales from schizophrenia patients and control subjects than the standard time-resolved functional connectivity approaches. They also find significant differences between schizophrenia patients and control subjects at the smallest scale.

I find the theoretical framework extremely intriguing, and the paper well written, but I feel that there are important points to be addressed that prevent me from recommending publication at this stage. Please find below a list of concerns.

- The authors claim (Fig 3: Caption and text: line 208) that in most cases the DSVAE model is significantly more reliable than the IDSVAE method. According to the R-squared plots, this is not true: in three cases DSVAE model is significantly more reliable than the IDSVAE method, in three cases IDSVAE method is significantly more reliable than the DSVAE model, and in three cases they are comparable. I would relax such a claim. There is a typo in Fig 3: the last plot of line three should be LS=8, CS=8

- It is not clear which is the dependence of the method proposed on the noise of time series and, in turn, on the preprocessing of data.

- It is not reported how multi-site data harmonization was performed

- The authors are using group-level analysis, but the neuroscience community is moving towards individual functional and structural analyses. What is the individual variability of this approach?

- Recently, a similar approach has been presented (Morioka et al. NeuroImage Volume 218, September 2020, 116989) to identify local temporal structures, referred to as temporal primitives, reflecting dynamic resting-state networks. Can the authors explain the differences and analogies between the two methods and where the new one presented in this manuscript shows to be better?

6. PLOS authors have the option to publish the peer review history of their article (what does this mean?). If published, this will include your full peer review and any attached files.

Reviewer #1: No

Reviewer #2: **Yes: **Tommaso Gili

---

## [Author Response · Author response to Decision Letter 1]

4 Feb 2025

We want to thank the reviewers for their insightful comments, and to address them we have both edited the text in the manuscript and added new experiments. In brief, we have added significance tests to the classification and reliability experiments, and added two new appendices, testing 1) the effect of training-time noise on the classification and reliability of our models 2) whether using a CNN instead of an RNN to obtain context embeddings improves the classification performance of our model. We also completely refactored the code, improved hyperparameter tuning, reran all the experiments, and made the code available at https://github.com/eloygeenjaar/dynamical-motifs. In cases where our language in the initial version of the manuscript was unclear, we have added more in-depth explanations or analysis, and have extended our descriptions.

Reviewer 1

We want to thank the reviewer for their insightful comments, and for recognizing the comprehensive evaluation performed in our paper. We respond point-by-point to most of the reviewer’s comments below, and have revised our manuscript to address the comments.

Weaknesses

2.) While linear separability is an important metric, the heavy focus on it may overlook other potentially relevant aspects of the embeddings, such as non-linear relationships or other forms of separability that could be clinically significant.

We understand the reviewer’s concern that linear separability may not be the only way that embeddings could be separated from each other. The reason we specifically decided to evaluate our model using linear separability is twofold. First, it is common in the field of machine learning (linear probing in the field of self-supervised learning), especially representation learning to look specifically at linear separability as a way to evaluate how well each model can separate embeddings based on predetermined classes with its non-linearities. Specifically, since the neural networks themselves are non-linear, we would want them to find a space where embeddings are now linearly separable. If we require non-linearities to classify the embeddings, then that means the representation learning has not found the most ‘optimal’ embedding space. Second, all of the neural networks we use in this work are variational autoencoders (VAEs), which is generally assumed to create locally euclidean manifolds. This means that we can linearly interpolate between points on the manifold; this is a by-product of the reparameterization trick. Linearly separable embeddings thus make the model more interpretable since we can interpolate from one class to another in a linear fashion, and decode this linear interpolation trajectory to understand the features that the VAE learned to linearly separate the embeddings.

To clear this up, we have added the following sentences in the text:

“To test how well windows of fMRI activity from schizophrenia patients and control subjects are linearly separable in a latent space, we compare our proposed model to two baseline models as well as the widely used wFNC approach. The two baseline models we use are a model that only uses local embeddings (LVAE), and a model that only uses context embeddings (CVAE). The latter model has been proposed in previous work~\cite{zhang2021spatiotemporal}, and we follow their design choices by using a convolutional encoder and decoder. The LVAE model is the same as our model, but only uses local embeddings and no context embeddings. To obtain context embeddings from the LVAE model, we average the local embeddings over time in a window. To obtain embeddings of different sizes for wFNC, we use PCA on the training and validation set and then transform the test set into the same PCA space. To perform the classification, we concatenate the context embeddings from the training and validation set and train a linear support vector machine to separate context embeddings from schizophrenia subjects and control subjects in the latent space. The classification accuracy is calculated using the context embeddings from subjects in the test set. The reason we specifically chose to use a linear evaluation is because the latent space of variational autoencoders (VAEs) is often assumed to be locally Euclidean. This enables linear interpolation between points in the latent space to be meaningful when decoded into the original data space. Since our models are trained with a variational loss, and because linear separation makes our exploration of group differences more interpretable in Sections Cluster analysis & Cluster visualization, we decided to focus on linear separability. The results are shown in Fig 2. Note, we compare models with different local and context sizes to verify results are not spurious with respect to the chosen local and context embedding sizes.”

3.) While the study links embedding clusters to demographic variables and cognitive scores, the clinical interpretation of these findings is not fully explored. There is a gap between the technical findings and their practical application in clinical decision-making, which could limit the study’s real-world impact.

We agree that our manuscript did not appropriately describe how our findings/model could lead to real-world impact. To address this concern, we have added the following paragraph to the discussion.

“It is important to note that our findings also indicate specific windows from the fMRI timeseries where certain connectivity patterns are either higher or lower with respect to the average wFNC across all subjects diagnosed with schizophrenia. Since our findings indicate that subjects diagnosed with schizophrenia who are older and have lower CMINDs scores spend significantly more time in these connectivity patterns. Especially given the significantly lower CMINDs scores, these dynamic patterns and how often they occur during a resting-fMRI scan indicate an increasing severity of symptoms, and as mentioned above, negative symptoms specifically. Since the identified patterns of connectivity are specific to certain areas of the brain, stimulation in those areas could potentially help disrupt the dynamical patterns we have identified as being linked to schizophrenia. With the effectiveness of transcranial magnetic stimulation (TMS) in treating negative symptoms for patients diagnosed with schizophrenia [1], our findings indicate potentially fruitful stimulation sites that relate to symptom severity.”

[1] Lorentzen, R., Nguyen, T. D., McGirr, A., Hieronymus, F., & Østergaard, S. D. (2022). The efficacy of transcranial magnetic stimulation (TMS) for negative symptoms in schizophrenia: a systematic review and meta-analysis. Schizophrenia, 8(1), 35.

4.) The study primarily focuses on varying local and context sizes but does not extensively explore other variations in model architecture or hyperparameters that might influence performance, such as different types of encoders/decoders or alternate methods for embedding aggregation.

Although many of the results in the paper show varying local and context sizes, we do perform an extensive hyperparameter exploration for each model. We train around 128 different versions of each model and choose the model with the average (across seeds) validation loss as the final model for each local and context size. This constitutes training around 1152 models for just the DSVAE results, and 1152 for the IDSVAE model. Since the DSVAE and IDSVAE perform different ways of learning the context embeddings, we do believe we are looking at alternate versions of embedding aggregation, unless we misunderstand the reviewer’s point. Although it is hard to change the decoder in the DSVAE/IDSVAE models, since decoding happens for each timestep independently, we have added new results for a DSVAE/IDSVAE model that uses a 1D convolutional encoder for the context embeddings. These results are discussed in Appendix D, but we do not find the convolutional encoder to significantly improve over the RNN encoder. In fact, when LS=2,CS=8 the DSVAE model with an RNN is significantly better. Moreover, we have added the following sentences to our methodology section to increase the clarity of our hyperparameter search approach:

“The necessary code to reproduce the proposed model is available on [https://github.com/eloygeenjaar/dynamical-motifs](Github). Each model is trained with PyTorch 2.0~\cite{Ansel_PyTorch_2_Faster_2024} across $32$ different hyperparameters with $4$ seeds $[42, 1337, 1212, 9999]$ ($128$ runs total), and the hyperparameters with the best validation performance on average across the seeds is used in the final evaluation of each model. The hyperparameter ranges for each model are described in Appendix A. Each model is trained with an NVIDIA GeForce RTX 1080 or A40.”

Comments to authors

1.) The trade-off between classification accuracy and computational reproducibility is a key issue highlighted in the paper. The IDSVAE model performs better in some cases but has lower reliability, suggesting a need for further refinement to balance these two aspects effectively.

To address this comment we have improved our hyperparameter search for each model. Initially we were selecting the best hyperparameters for each random seed independently. This means that the model is different for each seed, e.g. it has a different hidden dimension for the MLP in the encoder for each random seed. Instead, in our updated framework, we select the best hyperparameter setting for each model by averaging the validation loss across the four random seeds. This ensures that the same hyperparameter setting is used for each random seed as described above. To make our discussion of the models more rigorous we have added significance tests and now discuss the classification vs reliability trade-off as follows with these new significance results:

“In Fig 3 we can see two main trends. First, that the DSVAE model performs the best when the local size and context sizes are smallest: (LS=2,CS=2 $p<0.001$, LS=2,CS=4 $p<0.0001$, and LS=4,CS=2 $p<0.05$), and that the IDSVAE is more reliable with a larger local size (LS=8,CS=2 $p<0.001$ and LS=8,CS=8 $p<0.0001$). Moreover, larger context dimensionalities generally lead to worse reliability. This is largely expected since higher-dimensional spaces are essentially 'bigger', and there are thus more variations of embedding spaces that can lead to a good generative model of the data. Furthermore, in the cases where the DSVAE model is better than the IDSVAE model, its reliability is fairly high overall. Especially the LS=2,CS=4 model has a high classification accuracy, see Fig 2, good computational reproducibility, see Fig 3, and important for the next sections, is easy to visualize because the context embeddings are 2-dimensional. Hence, we will use this model for further analysis in the upcoming sections.”

2.) The manifold comparison shows that the proposed model's embeddings do not closely match wFNC features, with a low R-squared score of 0.19. While this indicates that the new embeddings capture different features, the clinical or biological significance of these differences is not well explained.

We understand that our current explanation of this analysis is currently confusing. Our main goal with the manifold comparison between the wFNC features and our model’s context embeddings was to show that our model provides complementary features to the wFNC features. In terms of clinical significance this means that there are two sets of quantitative features that can be used to make decisions by clinicians. To explain this idea further we have added the following text to the results section:

“The R-squared score is 0.14, this indicates that our model does not learn features that are essentially similar to connectivity features. Our model thus learns unique dynamical features and can provide complementary quantitative information about a window of fMRI data to practitioners. We additionally provide a visualization of wFNC features based on our model's embedding space in Appendix C to show differences in representations.”

Lastly, we do want to note that in the Cluster visualization section we describe the biological differences that our model captures and link them to previous findings for schizophrenia in the Discussion section.

3.) The visualization of clusters and embeddings in two dimensions is a strength, but the interpretation of these visualizations needs to be expanded. More detailed explanations and additional metrics must be provided to link the visual patterns to specific clinical or cognitive outcomes.

To increase the interpretation of the visualizations, we have adapted our discussion and more specifically link the findings from the most separated cluster to clinically applicable consequences. The updated text in the discussion is as follows:

“Lastly, the most separated cluster shows decreased visual-sensorimotor, decreased subcortical-sensorimotor, and increased visual-subcortical domains. These increases and decreases are with respect to the rest of the schizophrenia patients, and thus do not reflect a decrease or increase with respect to control subjects necessarily. Previous work has shown that reduced visuo-motor could be prospectively related to social deficits in schizophrenia patients [1]. The patterns observed in the most separated cluster in our analysis, which shows reduced visual-sensorimotor connectivity, may thus be indicative of negative symptoms. Additionally, decreased cortical-subcortical motor loop interaction has been related to hypokinesia, a general slowing of movement and a negative symptom, in schizophrenia patients [2, 3]. This aligns with the decreased subcortical-sensorimotor connectivity we find in the most separated cluster, and underlines its potential indication as a severity of negative schizophrenia symptoms.

It is important to note that our findings also indicate specific windows from the fMRI timeseries where certain connectivity patterns are either higher or lower with respect to the average wFNC across all subjects diagnosed with schizophrenia. Since our findings indicate that subjects diagnosed with schizophrenia who are older and have lower CMINDs scores spend significantly more time in these connectivity patterns. Especially given the significantly lower CMINDs scores, these dynamic patterns and how often they occur during a resting-fMRI scan indicate an increasing severity of symptoms, and as mentioned above, negative symptoms specifically. Since the identified patterns of connectivity are specific to certain areas of the brain, stimulation in those areas could potentially help disrupt the dynamical patterns we have identified as being linked to schizophrenia. With the effectiveness of transcranial magnetic stimulation (TMS) in treating negative symptoms for patients diagnosed with schizophrenia [4], our findings indicate potentially fruitful stimulation sites that relate to symptom severity.”

[1] Lu, P. Y., Huang, Y. L., Huang, P. C., Liu, Y. C., Wei, S. Y., Hsu, W. Y., ... & Tseng, H. H. (2021). Association of visual motor processing and social cognition in schizophrenia. npj Schizophrenia, 7(1), 21.

[2] Walther, S. (2015). Psychomotor symptoms of schizophrenia map on the cerebral motor circuit. Psychiatry Research: Neuroimaging, 233(3), 293-298.

[3] Menon, V., Anagnoson, R. T., Glover, G. H., & Pfefferbaum, A. (2001). Functional magnetic resonance imaging evidence for disrupted basal ganglia function in schizophrenia. American Journal of Psychiatry, 158(4), 646-649.

[4] Lorentzen, R., Nguyen, T. D., McGirr, A., Hieronymus, F., & Østergaard, S. D. (2022). The efficacy of transcranial magnetic stimulation (TMS) for negative symptoms in schizophrenia: a systematic review and meta-analysis. Schizophrenia, 8(1), 35.

4.) The analysis of distance traveled by embeddings between time steps reveals some differences between schizophrenia patients and controls. However, this observation needs to be discussed in detail to understand the underlying dynamics or to correlate these movements with specific neural or behavioral phenomena.

After re-running all the experiments with our updated and improved codebase we found that these results do not replicate well, so we have removed them from the revised m

---

## [Editor Report · Decision Letter 1]

Providing context: Extracting non-linear and dynamic temporal motifs from brain activity

PONE-D-24-29597R1

Dear Dr. Geenjaar,

We’re pleased to inform you that your manuscript has been judged scientifically suitable for publication and will be formally accepted for publication once it meets all outstanding technical requirements.

Kind regards,

Federico Giove, PhD

Academic Editor

PLOS ONE
---

## [Editor Report · Acceptance letter]

PONE-D-24-29597R1

PLOS ONE

Dear Dr. Geenjaar,

I'm pleased to inform you that your manuscript has been deemed suitable for publication in PLOS ONE. Congratulations! Your manuscript is now being handed over to our production team.

Kind regards,

on behalf of

Dr. Federico Giove

Academic Editor

PLOS ONE